# Parietal Epithelial Cell Behavior and Its Modulation by microRNA-193a

**DOI:** 10.3390/biom13020266

**Published:** 2023-01-31

**Authors:** Joyita Bharati, Praveen N. Chander, Pravin C. Singhal

**Affiliations:** 1Institute of Molecular Medicine, Feinstein Institute for Medical Research and Department of Medicine, Zucker School of Medicine at Hofstra-Northwell, Hempstead, NY 11549, USA; 2Department of Nephrology, Post Graduate Institute of Medical Education and Research, Chandigarh 160012, India; 3New York Medical College, Touro College and University System Valhalla, Valhalla, NY 10595, USA

**Keywords:** parietal epithelial cell, podocyte, micro-RNA, FSGS, crescentic glomerulonephritis

## Abstract

Glomerular parietal epithelial cells (PECs) have been increasingly recognized to have crucial functions. Lineage tracking in animal models showed the expression of a podocyte phenotype by PECs during normal glomerular growth and after acute podocyte injury, suggesting a reparative role of PECs. Conversely, activated PECs are speculated to be pathogenic and comprise extracapillary proliferation in focal segmental glomerulosclerosis (FSGS) and crescentic glomerulonephritis (CrescGN). The reparative and pathogenic roles of PECs seem to represent two sides of PEC behavior directed by the local milieu and mediators. Recent studies suggest microRNA-193a (miR193a) is involved in the pathogenesis of FSGS and CrescGN. In a mouse model of primary FSGS, the induction of miR193a caused the downregulation of Wilms’ tumor protein, leading to the dedifferentiation of podocytes. On the other hand, the inhibition of miR193a resulted in reduced crescent lesions in a mouse model of CrescGN. Interestingly, in vitro studies report that the downregulation of miR193a induces trans-differentiation of PECs into a podocyte phenotype. This narrative review highlights the critical role of PEC behavior in health and during disease and its modulation by miR193a.

## 1. Introduction

Parietal epithelial cells (PECs) are resident kidney cells that line the inner Bowman’s capsule [1]. They originate from the metanephric mesenchyme and undergo a mesenchymal-to-epithelial transition during early nephrogenesis, similar to podocytes. Recently, in the 21st century, increasing evidence suggests that PECs have crucial functions in health and disease [2,3,4,5]. Podocyte loss is known to drive glomerulosclerosis [6], a unifying mechanism for end-stage kidney disease in all glomerular diseases. In addition to podocyte loss, PEC activation is now known to be associated with severe glomerular diseases. Lineage tracking research has shown that PECs comprise extracapillary proliferation in animal models of podocytopathies and crescentic glomerulonephritis (CrescGN) [7,8,9,10,11]. Activated PECs are shown to participate in focal segmental glomerulosclerosis (FSGS) lesions [12], the most common primary glomerulopathy leading to end-stage kidney disease [13]. PEC activation may link podocyte loss and progressive glomerulosclerosis. Although the evidence on this pathogenic role of PECs [7,8,9,10,11,12] seems contradictory to their proposed reparative role [2,3,4,5], these may represent two sides of PEC behavior based on increased or decreased adverse or conducive milieus and mediators. Recent research on regulators of gene expression has signaled the role of microRNA-193a (miR193a) as a mediator in the pathogenesis of FSGS and CrescGN [14,15,16,17,18]. The modulation of miR193a has been reported to alter PEC behavior in vitro [16]. Further research to understand the effect of miR193a or its downstream mediators on PECs, especially animal models, will provide insight into this connection. 

This narrative highlights the evidence of PEC behavior in glomerular diseases (FSGS and CrescGN) and its modulation by miR193a.

## 2. PECs and Their Phenotypes

PECs are attenuated squamous cells with a condensed nucleus and express specific markers such as paired box gene (PAX) 2, PAX8, annexin A3, Claudin-1/2/16, and Ubiquitin C-terminal hydrolase-L1 (UCH-L1) [8]. Unlike podocytes, PECs proliferate at a low level throughout life [19].

### 2.1. Types of PECs in Human Kidneys

#### 2.1.1. Based on Location and Expression of Molecular Markers

Bariety et al. [20] used the term “parietal podocytes” for the PECs at the vascular pole of glomeruli that co-express PAX2 (PEC marker) and WT1 (+other podocyte markers) and “other PECs” for those located away from the vascular pole that expressed cytokeratin without podocyte proteins. The origin and function of parietal podocytes are not yet clear. They are similar to visceral podocytes in every aspect except the presence of PAX2, which implies their ability to divide. Ronconi et al. [3] have classified PECs based on stem cell markers (CD133/CD24) and podocyte marker podocalyxin (PDX) expression: the PECs at the vascular pole (CD133-CD24-PDX+) are differentiated cells resembling podocytes, the PECs in between the vascular and tubular poles (CD133+CD24+PDX+) are transitional cells, i.e., podocyte-committed progenitors with a low regenerative potential, and the PECs at the tubular pole are progenitor cells (also called “adult parietal multipotent progenitors” expressing CD133+CD24+PDX-) with a high regenerative potential. The progenitor cells at the tubular pole express only the stem cell markers and not the podocyte markers, however, they were capable of transforming into both podocytes and proximal tubular cells in vitro and in vivo. Hence, they were called bipotent progenitors. These bipotent progenitor cells were shown to reduce albuminuria and glomerular damage when injected into diseased mice (Figure 1).

#### 2.1.2. Based on Morphology

Kuppe et al. [21] have described PECs as flat, intermediate, and cuboidal. Cuboidal PECs have brush border-mimicking proximal tubular cells and line the Bowman’s capsule over the tubular pole. Intermediate PECs express the phenotype of injured proximal tubule cells that are trying to regenerate, i.e., they do not have a brush border, are triangular in shape with less cytoplasm, and express markers such as keratin 7 and keratin 19. They are seen at the junction between the cuboidal PECs and flat PECs in glomerular disease (Figure 1). Intermediate and cuboidal PECs had a higher activation potential, i.e., they preferentially express CD44 and Ki-67, and were shown to be involved in sclerosis/crescent formation in mouse models of FSGS and CrescGN, respectively, whereas flat PECs had lower activation potential. Intermediate PECs were shown to be the predominant PEC subtype forming synechiae between the glomerular tuft and Bowman’s capsule in human kidney biopsies of early FSGS recurrences that predominantly had tip lesions. Intermediate PECs were less and flat PECs were more abundant in kidney biopsies with FSGS not otherwise specified (NOS). Whether the abundance of intermediate PECs in tip lesion and the abundance of flat PECs in FSGS-NOS reflects different stages of injury and results in favorable prognosis of tip lesion remains to be studied.

## 3. Are PECs Reparative?

Podocytes are lost in the urine under normal physiological conditions [5]. The continuous replacement of the lost podocytes possibly prevents proteinuria from developing. Clinical reports of the reversal of diabetic nephropathy in humans [22] and animal models [23,24] suggest podocyte regeneration under diseased conditions. PECs were observed to acquire podocyte phenotype and markers when cultured in conducive media containing vitamin D3, retinoic acid, and dexamethasone [3]. The evidence on “Are PECs the source of podocyte renewal during growth and during disease?” is discussed in the following section.

### 3.1. Are PECs Progenitors for Podocytes during Postnatal Glomerular Growth?

PECs and podocytes share a common progenitor [25]. Puelles et al. [26] observed healthy human adults with larger glomeruli to have more podocytes than children, reflecting podocyte gain after birth. They noted the PECs co-expressing stem cell markers (CD24, CD133) and podocyte markers (p57, synaptopodin) near the vascular pole in both human children (≤3 years) and young adult glomeruli. Appel et al. [27] observed the PECs to replace the podocytes in adolescent mice during growth. The progenitor PECs express stem cell markers CD133 and CD24. The trans-differentiation of PECs to podocytes was shown using lineage tracking of PECs in rats followed postnatally on day 5 or 10 when nephrogenesis was complete. The PECs were observed to be increased over the glomerular tuft and all the PECs co-expressed podocyte markers (podocin, nephrin, and p57) along with the PEC activation marker (CD44) [27]. Berger et al. [28] showed recruitment of podocytes from PECs during normal growth until infancy in mice and in a limited number of human kidney biopsies. These studies suggest that podocytes are generated postnatally and that PECs are the likely source in animal models; however, it is still not clear if that holds true in humans.

### 3.2. Are PECs Progenitors for Podocytes during Aging? 

Podocyte regeneration was not observed in aging mice kidneys, which showed progressive loss of podocytes and the development of glomerulosclerosis [29]. Similarly, Berger et al. did not observe podocyte renewal from PECs in aging mice [27]. Nevertheless, Kaverina et al. have recently documented PEC migration and differentiation to podocytes in glomeruli with low podocyte count in otherwise healthy but aged mice [30]. It is not yet known if PECs that differentiate into podocytes can function as healthy podocytes and reverse aging-induced kidney changes. 

### 3.3. Are PECs Progenitors for Podocytes during Kidney Disease?

In a mouse model of partial acute podocyte depletion, genetically labeled PECs were observed to give rise to differentiated cells resembling podocytes after four weeks [29]. This finding coincided with proteinuria reduction in the mice [29]. In a mouse model of FSGS caused by acute podocyte depletion, genetically labeled PECs were shown to migrate to the affected glomerular tuft [31]. On day 28, the same PECs were noted to co-express podocyte markers, correlating with increased podocyte number and reduced glomerulosclerosis severity. The authors later reported [32] that PECs started increasingly co-expressing podocyte markers on day 28 and day 56 after injury. Lasagni et al. [33] noted that renal progenitor cells lining Bowman’s capsule differentiate into podocytes through intraperitoneal retinoic acid, leading to disease remission in a mouse model of FSGS. Ronconi et al. [3] showed that injection of progenitor PECs in mice with adriamycin-induced nephropathy reduced albuminuria and chronic glomerular damage. 

While Eng and colleagues [31,32] provide strong evidence of the ability of PECs to differentiate into podocytes in adult mice with acute podocyte depletion, contemporary researchers argue that the complexity of transgenic tagging of PECs and podocytes might have confounded their analysis [34]. Berger et al. [28] noted that PECs did not differentiate into podocytes in response to nephron loss induced by progressive partial nephrectomies in mice. Similarly, Wanner et al. [29] observed that nephron loss causing glomerular hypertrophy was not followed by podocyte renewal. Therefore, PECs did not contribute to podocyte regeneration after gradual nephron loss. 

The source of podocyte renewal during the recovery of human glomerular diseases is not confirmed to be PECs. Therefore, evidence on the reparative role of PECs after podocyte injury remains controversial. 

## 4. Are PECs Pathogenic?

### 4.1. PEC Activation in Classical FSGS and Collapsing Glomerulopathy

The PECs respond rapidly to extensive podocyte loss by activating themselves to migrate and secrete an extracellular matrix to seal the injured GBM. In an experimental mouse model, the severity of podocyte loss was correlated with the degree of proliferation of PECs [35]. Although various molecules are associated with PEC activation, no single cellular process has been confirmed to drive their trigger [36]. Activated PECs express CD44 and secrete the extracellular matrix of the Bowman’s capsule type, which is different from that present in the GBM. Excessive PEC activation is associated with the progression of glomerular diseases [7,8,9,10,11]. In genetically labeled PEC reporter mice with FSGS, the migration of PECs towards the injured glomerular tuft was noted, and the intensity of CD44 expression by PECs was positively related to the severity of glomerulosclerosis [27,31]. Furthermore, activated PECs formed the cellular adhesions between the Bowman’s capsule and the glomerular tuft in human FSGS [21]. Increased CD44 expression by PECs on the glomerular tuft has been shown to correlate with poor clinical kidney function in children with FSGS [37]. Benigni et al. [38] reported the ability of angiotensin-converting enzyme inhibitors (ACEi) to normalize glomerular lesions by blocking PEC activation in a rat model of glomerulosclerosis. These studies suggest that PEC activation, marked by CD44 expression, is associated with more severe glomerulosclerosis in FSGS. 

The degree of PEC proliferation correlates with the severity of podocyte injury [35]. The massive (perhaps acute) loss of podocytes can invoke extensive PEC proliferation, resulting in the formation of layers of cells (pseudo-crescents) and consequent glomerular tuft collapse, termed collapsing glomerulopathy (CG) [39]. While its categorization within the ambit of FSGS is controversial [40,41], typical CG is associated with podocyte loss and massive albuminuria. CG typically occurs in the setting of the human immunodeficiency virus (HIV) infection in patients with APOL1 high-risk alleles (G1 and G2) [42]. The normal APOL1G0 allele is expressed in podocytes but not PECs [18] and is speculated to be protective against HIV-associated nephropathy [43,44]. APOL1G0 facilitates PEC transition to podocytes [18] in vitro, and its association with miR193a is described below. Interestingly, HIV has been demonstrated to induce APOL1G0 expression in PECs and, therefore, increase PEC transition to podocytes in vitro [18]. In contrast, HIV infection is also known to enhance the expression of APOL1 risk alleles [45] and trigger FSGS development by causing podocyte de-differentiation in vivo [46]. Moreover, APOL1 risk alleles are observed to cause altered podocyte function [47]. Thus, CG likely develops in HIV positive patients carrying APOL1 risk alleles when the following events occur simultaneously: (i). massive podocyte loss (direct HIV-induced and APOL1G1/G2 risk allele-induced), (ii). PEC activation following podocyte injury, and (iii). compromised podocyte renewal by PECs (due to lack of APOL1G0). On the other hand, in the pathogenesis of classical FSGS, only the loss of a critical number of podocytes acts as the trigger for activating adjacent PECs that, based on the severity of podocyte injury and the local environment, result in either progressive sclerosis (deposition of excessive extracellular matrix) or the repair of the glomerular filtration barrier (podocyte renewal and sealing of the denudation).

### 4.2. PEC Activation in Crescentic GN

CrescGN is characterized by extracapillary proliferation in the Bowman’s space in response to a breach in GBM and the resultant flow of inflammatory mediators, such as immune cells and phlogogenic plasma [10]. Activated PECs are the predominant proliferating cells in the mouse nephrotoxic nephritis model of inflammatory CrescGN [7,36]. These activated PECs are a population of CD133+CD24+ progenitor cells, as shown in human kidney biopsies [11]. De novo CD9 expression by PECs is a prerequisite for CD44 expression and activation of PECs [48]. C-X-C Motif Chemokine Receptor 4 (CXCR4) and angiotensin II/angiotensin II type-1 (AT-1) receptor pathways are reported to be instrumental in activating PECs to form crescents [49,50]. Treatment with ACEi resulted in the normalization of CXCR4 staining in the animal model of CrescGN [49]. Further, CXCR2 and its ligand are expressed in PECs and podocytes, respectively, in experimental FSGS [51]. Therefore, activated PECs, characterized by chemokine receptor expression, and podocytes, marked by chemokine receptor ligand expression, appear interrelated. Hence, podocyte injury could trigger PEC activation in CrescGN, similar to that seen in FSGS. 

Although activated PECs in both human FSGS and CrescGN show a similar staining pattern, CD44, claudin-1, CD24, and CD133 [36], the manner of PEC activation is different. Whereas adhesions of glomerular capillary tufts to the Bowman’s Capsule are early features in FSGS [21], they are not prerequisites for forming cellular crescents [52]. PECs are mainly proliferative in CrescGN in response to an inflammatory stimulus after GBM rupture. However, PECs primarily acquire a migratory and secretory phenotype in FSGS in response to podocyte loss. Interestingly, CG shares its phenotype with sclerotic lesions in FSGS and extracapillary proliferation in CrescGN (Figure 2). They are differentiated based on the phenotype of the adjacent glomeruli in the same section, a breach of the GBM and presence of immune cells in the Bowman’s space in CrescGN, and a collapse of the capillary loops without breach of GBM and absence of immune cells in CG. Nonetheless, distinguishing severe CG with circumferential pseudo-crescent formation from severe CrescGN with compression of the glomerular tuft (resembling collapse) can sometimes be difficult on morphology alone. The molecular basis of PEC activation in these diseases might be different. 

## 5. MicroRNA-193a and PEC Response 

PEC activation follows direct podocyte injury in FSGS and indirect podocyte loss in CrescGN [12,36]. Multiple mediators link podocyte loss to PEC activation [36]. In line with this, miR193a, a new mediator linking podocyte loss and PEC activation, has gained attention for its differential effects on podocytes and PECs [14,15,16,17,18] (Figure 3).

MicroRNAs are small (approximately 20–25 nucleotides) non-coding RNAs with a predominant inhibitory role in gene expression as they destabilize messenger RNAs [52]. MicroRNAs are stable in urine and blood and they are reported to be promising biomarkers in diagnosing CKD with a slightly higher accuracy of urine than blood samples [53]. Specific microRNA profiles are associated with acute kidney injury, allograft rejection, diabetic nephropathy, and glomerular diseases [54]. Similarly, specific microRNA profiles are associated with cellular processes such as fibrosis, inflammation, and the epithelial–mesenchymal transition. Some microRNAs are protective, e.g., miR30 and miR146a in diabetic nephropathy, whereas others are associated with the progression of disease, e.g., miR193a and miR92a in glomerular diseases [54]. One microRNA can modulate the expression of multiple target genes. Long non-coding RNAs are another type of non-coding RNAs that can interrupt the function of microRNAs and their interaction has been shown to result in the final regulatory effect of microRNAs. Therefore, before extensive evaluation of the role of microRNAs as biomarkers, factors such as non-specificity for a particular cellular pathway and regulation by other molecules need to be studied in depth. miR193a has two arms: dominant arm (miR193a-3p) and a passenger arm (miR193a-5p). They were originally discovered in the context of tumorigenesis as they have a tumor suppressor effect, for e.g., in gastric cancer [55], colorectal cancer [56,57], medulloblastoma [58], epithelial ovarian cancer [59], and prostate cancer [60]. miR193a is expressed normally in the kidneys and is associated with FSGS and experimental crescentic GN. Interestingly, the connection between miR193a and kidneys was discovered serendipitously when miR193a transgenic mice used to study breast tumorigenesis were noted to develop albuminuria and fatal kidney failure with scarred kidneys [14,15]. Real time polymerase chain reaction is commonly used to estimate microRNAs in blood and urine.

In normal human kidneys, miR193a is predominantly expressed in PECs [14]. miR193a is a negative regulator of Wilms’ tumor protein 1 (WT-1) [14]. WT-1 is a marker of mature podocytes and a crucial transcription factor converting mesenchymal progenitors towards podocyte phenotype during nephrogenesis [61]. WT1 also forms a repressor complex on the promoter site of the PAX2 gene and causes downregulation of the PEC phenotype [17]; therefore, the downregulation of WT1 from elevated miR193a in precursor progenitor cells likely enhances the expression of PAX2 and PEC phenotype. In a nutshell, the expression levels of miR193a by precursor progenitor cells determine (through modulation of WT1) their net phenotype (PECs vs. podocytes). In mature podocytes, an upregulation of miR193a lowers WT-1, followed by podocyte foot process effacement [14]. miR193a transgenic mice exhibited downregulation of WT-1, complete foot process effacement in podocytes, and the development of proteinuria after four weeks of miR193a induction. The glomerular tuft displayed FSGS within the next 12 weeks, and the inhibition of miR193a led to reduced proteinuria [14]. Notably, although miR193a was overexpressed in the podocytes of biopsies of primary FSGS, genetic FSGS and minimal change disease exhibited low levels of miR193a, resembling normal kidneys. The transplantation of wild-type kidneys to an affected animal (miR193a transgenic mouse) did not lead to disease in the wild-type kidneys, suggesting that circulating factors were not implicated in the pathogenesis of FSGS in these mice. However, clinical studies have demonstrated high miR193a in primary FSGS, presumably caused by circulating permeability factors. In children with nephrotic syndrome, urinary exosomal miR-193a levels were higher in those with FSGS than those with minimal change disease. Higher urinary exosomal miR193a level was directly proportional to the severity of glomerulosclerosis [62]. The destabilizing effect of miR193a on podocytes is consistent with its tumor-suppressive role in various solid cancers. The upregulation of miR193a, however, does not disrupt normal PEC structure, probably because their phenotype is dependent on it. Further, the downregulation of miR193a leads to the loss of PEC markers and the expression of podocyte markers within PECs [16]. Kietzmann et al. [16] used a cultured immortalized human PEC line to demonstrate the effect of miR193a on PEC differentiation into podocytes. miR193a upregulation was observed to induce podocyte dedifferentiation and miR193a downregulation-induced expression of podocyte markers within PECs [16]. The authors also noted enhanced expression of miR193 within the crescents in mouse models of CrescGN [16]. Larger clinical studies across different centers evaluating miR193a as a biomarker differentiating early FSGS from minimal change diseases are crucial for validation. Similarly, larger clinical studies evaluating the predictive ability of miR193a on kidney outcomes are needed.

In in vitro studies, media containing vitamin D3, retinoic acid, and dexamethasone attenuated miR193a expression in PECs and reduced PEC proliferation. Similarly, the inhibition of miR193 by genetic knockdown in mice resulted in the decreased crescent formation and proteinuria in a mouse model of crescent GN [16]. The mechanism of reduced proteinuria associated with crescent reduction upon miR193a knockdown was not precisely illustrated. While crescent reduction results in an improved glomerular filtration rate, it does not necessarily explain proteinuria reduction. A possible explanation could be that the downregulation of miR193a led to an environment favoring podocyte renewal that contributed to a decrease in proteinuria. Therapeutic agents specifically targeting miR193a in the kidneys to ameliorate FSGS and CrescGN need to be explored. Since the dosage of Vitamin D3 and retinoids required to down-regulate miR193a in vitro are likely to be associated with cellular toxicities in humans, these agents need to be modified to reduce adverse effects. Moreover, as vitamin D agonists regulate miR193a through the upregulation of the APOL1 gene, which includes its risk alleles, a cautious administration after APOL1 genotype profiling is crucial in patients with FSGS.

We previously showed that the downregulation of miR193a prevented podocyte dedifferentiation in a high glucose milieu [17] and enhanced podocyte marker expression by PECs under normal glucose conditions [18]. Both experimental conditions explored a possible axis connecting miR193a and APOL1G0 protein. In these in vitro studies, the downregulation of miR193a induced APOL1G0 expression in PECs. The use of a vitamin D agonist was shown to enhance the expression of APOL1G0 and decrease miR193a levels in podocytes treated with high glucose [17]. The downregulation of APOL1G0 enhanced miR193a levels, suggesting a bifunctional or bilateral relationship between APOL1(G0)-miR193a in the axis [63]. Further, the APOL1-miR193a axis was disrupted in differentiated podocytes expressing APOL1 risk alleles and was implicated in causing actin cytoskeleton disorganization [64]. The expression of APOL1 risk alleles in podocytes led to increased miR193a levels and favored podocyte de-differentiation [64]. Therefore, although APOL1 risk alleles exhibit toxic gain-of-function mutations, the effect on miR193a levels seem disparate from normal APOL1G0. The link between APOL1 renal risk alleles and miR193a needs to be studied further in vivo to understand the role of miR193a in causing or perpetuating APOL1 nephropathy. Additionally, the effect of enhancing APOLIG0 protein expression to reduce miR193a levels in glomerular diseases needs to be studied in vivo.

Bringing together the above studies on PEC response modulation by miR193a, we summarize the following highlights:

1. miR193a-induced podocyte injury is predominantly mediated through the downregulation of WT1 in podocytes, which is crucial for podocyte survival. Elevated miR193a exacerbates podocyte loss and that, in turn, causes excessive PEC activation. In addition, increased miR193a blocks PECs transitioning to podocytes. While increased miR193a in podocytes is observed in FSGS, the sequence of events leading to high miR193a is not yet known. Podocyte injury appears to drive PEC activation in both FSGS and experimental CrescGN, both of which are characterized by high miR193a in PECs limiting their transition to podocytes.

2. The downregulation of miR193a (by corticosteroids, retinoic acid, and vitamin D agonists) promotes the differentiation of PECs toward podocytes in vitro. The use of similar agents to downregulate miR193a, therefore, appears promising to improve podocyte renewal in diseased states, if successful strategies are employed to test such agents in clinical studies.

The following research questions are unanswered at present: (i). Does miR193a decide PEC behavior in different forms of glomerular injury in vivo? (ii). What causes an increase in miR193a expression in primary FSGS and CrescGN? (iii). Does the enhancement of APOLIG0 protein expression reduce miR193a levels in vivo?

While multiple aspects of the link between miR193a and PEC behavior in glomerular disease need to be explored, a crucial glaring question needs urgent attention—will the development of inhibitors of miR193a to treat kidney diseases be hindered by the promotion of cancer development in patients? Some of the future directions in this subject are discussed in Table 1.

## 6. Conclusions

PEC activation, besides podocyte loss, is a critical determinant of progressive glomerulosclerosis leading to chronic kidney disease. Activated PECs are the “pathogenic” side of PECs. On the contrary, PECs have a “reparative” role, i.e., they proliferate in response to podocyte loss and act as the source of podocyte renewal to sustain podocyte homeostasis. PECs are shown to transdifferentiate into a podocyte phenotype leading to podocyte gain during postnatal glomerular growth and to glomerular injury resolution after acute podocyte depletion in animal models. It remains to be studied if PECs are capable of postnatally replacing podocyte function after podocyte injury in humans. Therefore, research exploring the mediators upregulating the reparative role and downregulating the pathogenic role of the PECs seems promising in finding solutions to retard kidney disease progression. With the current progress in molecular research, microRNAs have gained considerable attention in regulating crucial cellular processes in common diseases. miR193a is expressed in PECs normally, probably because their phenotype is dependent on it. In vitro downregulation of miR193a leads to the loss of PEC markers and the expression of podocyte markers within PECs. High miR193a expression in PECs in experimental disease models favors their proliferation, and the inhibition of miR193a reduces PEC proliferation in vitro. The downregulation of miR193a and/or its downstream pathways might result in podocyte renewal and the prevention of progressive glomerulosclerosis. The modulation of the APOL1G0 and miR193a axis in the podocytes and PECs improves podocyte renewal from progenitor PECs in vitro. Further studies evaluating miR193a modulation on PEC behavior in vivo are needed to confirm the association in different forms of glomerular injury.

## Figures and Tables

**Figure 1 biomolecules-13-00266-f001:**
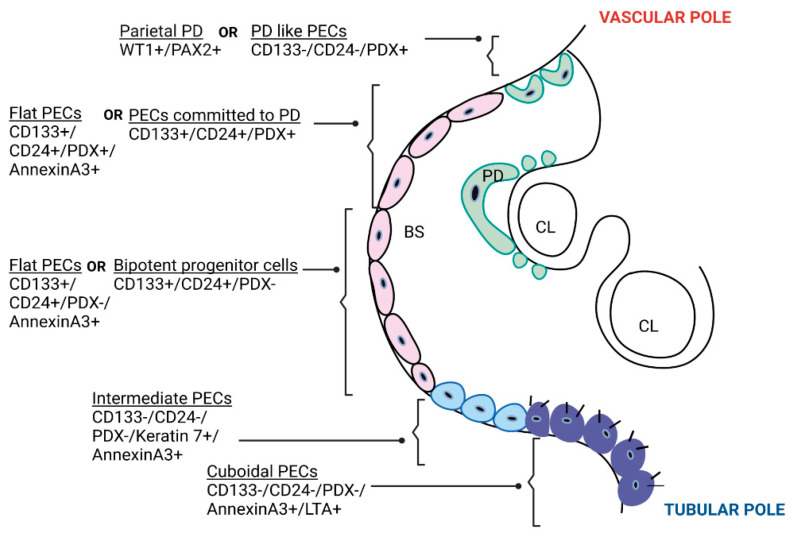
Types of parietal epithelial cells. A cartoon showing transverse section of a glomerulus showing types of PECs based on location, molecular phenotype [3,20] and morphology [21]. PECs based on stem cell marker (CD133/CD24) and PDX expression were located hierarchically along the Bowman’s capsule; differentiated cells resembling PD (CD133-CD24-PDX+) were located at the vascular pole, progenitor cells committed to PD (CD133+CD24+PDX+, also called “transitional cells”) were located between the vascular pole and tubular pole, and bipotent progenitor cells (CD133+CD24+PDX-, with a high regenerative capacity towards both PD and proximal tubular cells) were located at the tubular pole [3]. PECs based on morphology were: flat squamoid cells, intermediate PECs, and cuboidal PECs [21]. PD—podocyte; PECs—parietal epithelial cells; CL—capillary lumen; BS—Bowman’s space; WT1—Wilms’ tumor protein 1; PAX2—paired box gene 2; CD—cluster of differentiation; PDX—podocalyxin; LTA–lotus tetragonolobus agglutinin; FSGS–focal segmental glomerulosclerosis; CrescGN–crescentic glomerulonephritis.

**Figure 2 biomolecules-13-00266-f002:**
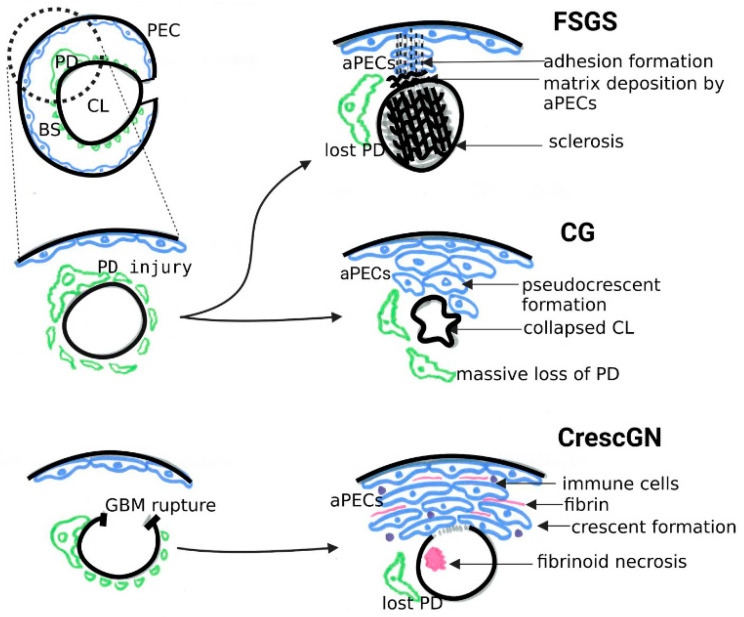
Schematic representation of pathogenesis of three diseases characterized by parietal epithelial cell activation- classical focal segmental glomerulosclerosis, collapsing glomerulopathy, and crescentic glomerulonephritis. FSGS—focal segmental glomerulosclerosis; CG—collapsing glomerulopathy; CrescGN—crescentic glomerulonephritis; PEC—parietal epithelial cell; PD—podocyte; GBM—glomerular basement membrane; BS—Bowman’s space; CL—capillary lumen; aPECs—activated parietal epithelial cells.

**Figure 3 biomolecules-13-00266-f003:**
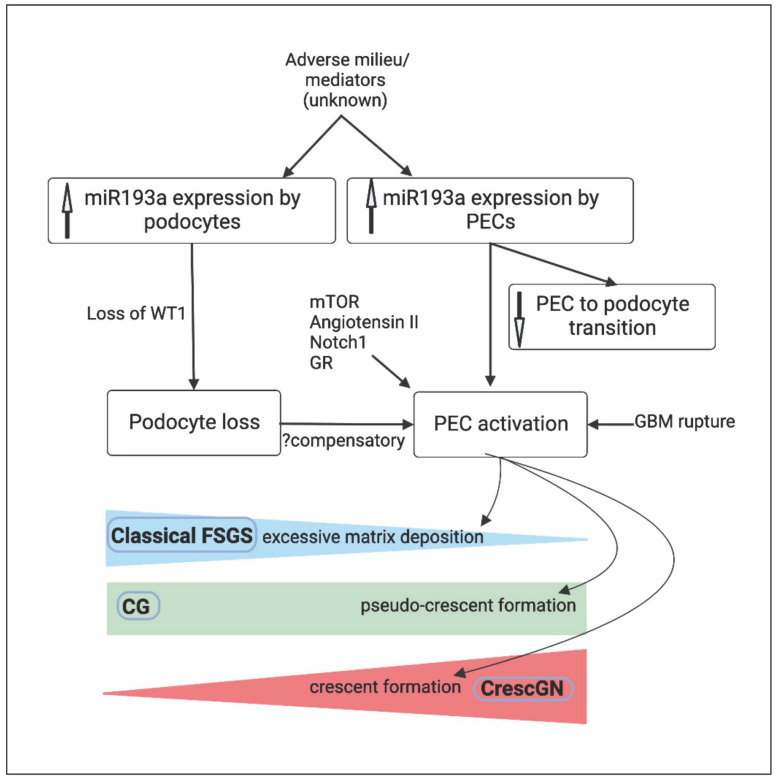
Algorithm summarizing mechanisms linking high miR193a, podocyte detachment, and PEC activation in classical FSGS (blue), CG (green), and CrescGN (red). PEC—parietal epithelial cell; PD—podocyte; WT1—Wilms’ tumor protein 1; BS—Bowman’s space; CL—capillary lumen; GBM—glomerular basement membrane; FSGS—focal segmental glomerulosclerosis; CG—collapsing glomerulopathy; CrescGN—crescentic glomerulonephritis; mTOR—mammalian target of rapamycin; Notch1—Neurogenic locus notch homolog protein 1; GR—glucocorticoid receptor.

**Table 1 biomolecules-13-00266-t001:** Future Research Perspectives.

Future Research Perspectives
Evaluation of PEC characteristics after miR193a inhibition in transgenic mice with miR193a-induced FSGS
Analysis of miR193a profile of the PECs and podocytes in animal models of crescentic GN during active disease and after resolution
Validation of miR193a profile in primary FSGS by other investigators
Comparison of miR193a profile in animals with transgenic expression of APOL1G0 and those without APOL1G0
Long-term tumorigenesis effects of miR193a inhibition in animal models

## Data Availability

Not applicable.

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
