# Peer review of "Parietal Epithelial Cell Behavior and Its Modulation by microRNA-193a"

_biomolecules, 2023, doi:10.3390/biom13020266_

Round 1

Reviewer 1 Report

The authors Bharati et al, wrote a systematic review on “Parietal epithelial cell behavior and its modulation by microRNA-193a” The authors updated the current knowledge on the role of microRNA-193a on parietal cells. The subject is novel and well-written. I suggest authors address the following comments.

Major Comments

1.     I suggest authors make a table for future perspectives and a conclusion section by highlighting the key findings about miRNA-193a and its role on parietal cells.

2.     There are grammatical, spelling, and spacing errors in the manuscript, I suggest authors correct them wherever applicable.

Author Response

Reviewer 1: The authors Bharati et al, wrote a systematic review on “Parietal epithelial cell behaviour and its modulation by microRNA-193a” The authors updated the current knowledge on the role of microRNA-193a on parietal cells. The subject is novel and well-written. I suggest authors address the following comments.

Major Comments

  1. I suggest authors make a table for future perspectives and a conclusion section by highlighting the key findings about miRNA-193a and its role on parietal cells.
  2. There are grammatical, spelling, and spacing errors in the manuscript, I suggest authors correct them wherever applicable.

Response 1: Thank you for highlighting these points. We understand the addition of these sections will make the manuscript explicit. We have addressed them by adding the Table 1 entitled “Future perspectives” and expanding the “conclusion section” on page #8.

Response 2: We have taken care in omitting any grammatical, writing, and formatting errors in the manuscript to the best of our abilities.

Reviewer 2 Report

This manuscript focuses on the behavior of parietal epithelial cells (PECs) in glomerular diseases focal segmental glomerulosclerosis (FSGS) and crescentic glomerulonephritis (CrescGN) and correlates PEC behavior with the miR193a expression level. Overall, the manuscript is comprehensive and provides the necessary evidence to support the expression level of miR193a and PEC activity and the severity of glomerulosclerosis. However, It would be informative to mention the role of miR193a in other tumor types like Medulloblastoma, Epithelial ovarian cancer, prostate cancer, and Gastric Cancer.1-4

1.      Northcott, P. A., Robinson, G. W., Kratz, C. P., Mabbott, D. J., Pomeroy, S. L., Clifford, S. C., ... & Pfister, S. M. (2019). Medulloblastoma. Nature reviews Disease primers5(1), 1-20.

2.      Zhang, S., Liu, J., He, J., & Yi, N. (2021). MicroRNA‑193a‑5p exerts a tumor suppressive role in epithelial ovarian cancer by modulating RBBP6. Molecular Medicine Reports24(2), 1-9.

3.      Yang, Z., Chen, J. S., Wen, J. K., Gao, H. T., Zheng, B., Qu, C. B., ... & Zhang, Y. (2017). Silencing of miR-193a-5p increases the chemosensitivity of prostate cancer cells to docetaxel. Journal of Experimental & Clinical Cancer Research36(1), 1-15.

4.     Chou, N. H., Lo, Y. H., Wang, K. C., Kang, C. H., Tsai, C. Y., & Tsai, K. W. (2018). MiR-193a-5p and-3p play a distinct role in gastric cancer: miR-193a-3p suppresses gastric cancer cell growth by targeting ETS1 and CCND1. Anticancer research, 38(6), 3309-3318.

Author Response

Reviewer 2: This manuscript focuses on the behaviour of parietal epithelial cells (PECs) in glomerular diseases focal segmental glomerulosclerosis (FSGS) and crescentic glomerulonephritis (CrescGN) and correlates PEC behaviour with the miR193a expression level. Overall, the manuscript is comprehensive and provides the necessary evidence to support the expression level of miR193a and PEC activity and the severity of glomerulosclerosis. However, It would be informative to mention the role of miR193a in other tumour types like Medulloblastoma, Epithelial ovarian cancer, prostate cancer, and Gastric Cancer.1-4 

  1. Northcott, P. A., Robinson, G. W., Kratz, C. P., Mabbott, D. J., Pomeroy, S. L., Clifford, S. C., ... & Pfister, S. M. (2019). Medulloblastoma. Nature reviews Disease primers5(1), 1-20.
  2. Zhang, S., Liu, J., He, J., & Yi, N. (2021). MicroRNA‑193a‑5p exerts a tumor suppressive role in epithelial ovarian cancer by modulating RBBP6. Molecular Medicine Reports24(2), 1-9.
  3. Yang, Z., Chen, J. S., Wen, J. K., Gao, H. T., Zheng, B., Qu, C. B., ... & Zhang, Y. (2017). Silencing of miR-193a-5p increases the chemosensitivity of prostate cancer cells to docetaxel. Journal of Experimental & Clinical Cancer Research36(1), 1-15.
  4. Chou, N. H., Lo, Y. H., Wang, K. C., Kang, C. H., Tsai, C. Y., & Tsai, K. W. (2018). MiR-193a-5p and-3p play a distinct role in gastric cancer: miR-193a-3p suppresses gastric cancer cell growth by targeting ETS1 and CCND1. Anticancer research, 38(6), 3309-3318.

Response: We thank the reviewer for the suggestion. We have added the above references in the manuscript in the “MicroRNA193a and PEC response” section on page #6.

Reviewer 3 Report

Overall it is a well-written review paper with illustrations for easier understanding. If the authors could propose experiments or studies for better elucidating the relationship between PEC activation and miR193a in disease progression to answer the listed question in the end, that would be better.   

Author Response

Reviewer 3: Overall it is a well-written review paper with illustrations for easier understanding. If the authors could propose experiments or studies for better elucidating the relationship between PEC activation and miR193a in disease progression to answer the listed question in the end, that would be better. 

Response: We thank the reviewer for highlighting this crucial point. We have addressed this in the section on “Future perspectives” which was added to list down possible experiments or studies for elucidating the relationship between PEC activation and miR193a in glomerular diseases. 

Round 2

Reviewer 1 Report

The authors Bharati et al sufficiently addressed the comments. Hence, the manuscript is suitable for publication.